# Unraveling the Epigenetic Landscape of Mature B Cell Neoplasia: Mechanisms, Biomarkers, and Therapeutic Opportunities

**DOI:** 10.3390/ijms26178132

**Published:** 2025-08-22

**Authors:** Nawar Maher, Francesca Maiellaro, Joseph Ghanej, Silvia Rasi, Riccardo Moia, Gianluca Gaidano

**Affiliations:** 1Division of Hematology, Department of Translational Medicine, Università del Piemonte Orientale and Azienda Ospedaliero-Universitaria di Alessandria, 15121 Alessandria, Italy; nawar.maher@uniupo.it; 2Division of Hematology, Department of Translational Medicine, Università del Piemonte Orientale and Azienda Ospedaliero-Universitaria Maggiore della Carità, 28100 Novara, Italy; francesca.maiellaro@uniupo.it (F.M.); joseph.ghanej@uniupo.it (J.G.); silvia.rasi@med.uniupo.it (S.R.); riccardo.moia@uniupo.it (R.M.)

**Keywords:** B cell malignancies, epigenetics, DNA methylation, histone modification, noncoding RNA

## Abstract

Epigenetic regulation is critical to B cell development, guiding gene expression via DNA methylation, histone modifications, chromatin remodeling, and noncoding RNAs. In mature B cell neoplasms, particularly diffuse large B cell lymphoma (DLBCL), follicular lymphoma (FL), and chronic lymphocytic leukemia (CLL), these mechanisms are frequently disrupted. Recurrent mutations in key epigenetic regulators such as *EZH2*, *KMT2D*, *CREBBP*, and *TET2* lead to altered chromatin states, repression of tumor suppressor genes, and enhanced oncogenic signaling. Dysregulation of specific microRNAs (e.g., miR-155, miR-21) further contributes to pathogenesis and therapeutic resistance. In DLBCL, hypermethylation of *SMAD1* and *CREBBP* mutations are associated with immune evasion and chemoresistance. In FL, *EZH2* gain-of-function and *KMT2D* loss-of-function mutations alter germinal center B cell programming, while in CLL, DNA hypomethylation patterns reflect the cell of origin and correlate with clinical outcome. Targeted therapies such as the EZH2 inhibitor tazemetostat have demonstrated efficacy in *EZH2*-mutant FL, while HDAC and BET inhibitors show variable responses across B cell malignancies. The limitations of current epigenetic therapies reflect the complexity of targeting epigenetic dysregulation rather than therapeutic futility. These challenges nonetheless highlight the relevance of epigenetic alterations as biomarkers and therapeutic targets, with potential to improve the management of mature B cell neoplasms.

## 1. Introduction

Mature B-cell neoplasms are a diverse group of lymphoid cancers arising from antigen-experienced B cells, typically at or beyond the germinal center stage [1,2]. They include malignancies derived from germinal center B cells (e.g., follicular lymphoma, FL and a fraction of diffuse large B cell lymphomas, DLBCLs), and post-GC B cells (e.g., certain DLBCLs), and memory B cells (e.g., chronic lymphocytic leukemia, CLL) [1,2]. Although these tumors often retain key immunophenotypic traits of their normal counterparts, they exhibit profound dysregulation of gene expression due to disruptions in the regulatory networks that control cellular identity and function [3]. This dysregulation is driven by both genetic and epigenetic alterations and leads to aberrant expression patterns, loss of normal differentiation, and malignant features such as uncontrolled growth and immune evasion [4]. For instance, the activated B cell-like (ABC) subtype of DLBCL is marked by constitutive activation of the NF-κB pathway, upon which these tumors depend for survival [5]. Similarly, certain DLBCL subsets rely on sustained BCL6 expression to repress its target genes, highlighting BCL6 as a central transcriptional driver. However, beyond transcription factors (TFs) like NF-κB and BCL6, gene expression is governed by the epigenetic landscape, that includes chromatin architecture shaped by DNA methylation, histone modifications, nucleosome remodeling and related mechanisms [6].

Epigenetics refers to heritable changes in gene function that occur without alterations to the DNA sequence [7]. These chromatin-based regulatory layers encode DNA sequence independent information that is essential for lineage fidelity, cell fate transitions, and the dynamic gene expression programs required for normal B cell development [4]. From naïve B cells to germinal center (GC) cells, and ultimately memory or plasma cells, each stage of B cell maturation demands precise coordination of transcriptional networks, signaling pathways and epigenetic machinery. The inherent plasticity of these processes contributes to the molecular heterogeneity seen in B cell malignancies. Notably, large-scale genomic analyses of follicular lymphoma (FL), DLBCL, and chronic lymphocytic leukemia (CLL), among others, consistently reveal a high mutational burden affecting chromatin modifiers, DNA methylation regulators, and TFs that disrupt normal epigenetic control [8].

Importantly, the recognition of epigenetic dysregulation as a key oncogenic driver has translated into therapeutic advances. Agents that target the epigenome, such as the DNA methylation inhibitors azacitidine and decitabine, have demonstrated significant clinical efficacy in myelodysplastic syndromes (MDS) and acute myeloid leukemia (AML) [9]. These successes provide a proof-of-principle that pharmacologic modulation of epigenetic alterations can produce meaningful therapeutic benefits and offer promising avenues for the treatment of mature B cell neoplasms.

As lymphomas often evolve from or hijack normal epigenetic programs, understanding how these regulatory mechanisms are subverted in a subtype-specific manner is essential. While several reviews have examined epigenetic mechanisms in B-cell malignancies, many are either disease-specific or focus on isolated pathways. Our review presents an integrated analysis across the three most prevalent mature B-cell lymphoid neoplasms, namely DLBCL, FL, and CLL selected for their shared lymphoid origin, overlapping epigenetic features, and the availability of comparable epigenomic data. We explore how aberrant DNA methylation, mutations in chromatin modifiers, and altered histone landscapes contribute to disease pathogenesis, and discuss how these alterations represent potential prognostic biomarkers that are guiding the development of novel epigenetic therapies. Furthermore, by integrating all epigenetic mechanisms with the most recent therapeutic strategies, our review may provide a framework for both research and clinical applications.

## 2. Epigenetic Mechanisms in B Cell Maturation

B cell maturation is a complex multi-step process strictly regulated by the interplay between different stage-specific TFs and epigenetic complexes, which define specific gene expression programs [10,11]. Epigenetic modifications play a crucial role in B cell maturation by using chemical modifications translated into instructions which can regulate gene expression and influence cell growth, apoptosis, development, differentiation, and immune response. The four main epigenetic modifications include DNA methylation, histone modifications, chromatin remodeling, and noncoding RNAs (Figure 1) [12,13].

### 2.1. DNA Methylation

DNA methylation involves the covalent addition of a methyl group to cytosines, mostly in the context of CpG dinucleotides. These CpG sites are often clustered in regions known as CpG islands, which are located in the promoters and/or first exons of approximately 60% of human genes [14]. The methylation machinery is represented by DNA methyltransferases (DNMT) (enzymes known as writers), 5-methylcytosine (5mC) binding proteins (readers), and by ten-eleven translocation methylcytosine dioxygenase (TET) enzymes (erasers), responsible for the demethylation of 5mC [15]. Methylation marks are related to enhancers and promoters, affecting heterochromatin and nuclear lamina associated domains, polycomb repressed regions and DNA repeats (Figure 1). The role of TET2 is sustained by Isocitrate Dehydrogenase 1 and 2 (IDH1/2) activity which consists of the conversion of isocitrate in α-ketoglutarate (α-KG), an important cofactor of TET2. Alterations of this enzyme lead to the formation of an oncometabolite taking part in the leukemogenesis process [16].

In the context of B cell maturation, the methylome represents a fundamental element in differentiation programs, activating or repressing cell pathways to define distinct features of each maturation step. Methylation effects profoundly depend upon their localization in the genome. If DNA methylation occurs in CpG rich gene promoters, it is associated with transcriptional silencing. Conversely, methylation of intragenic regions is more commonly linked to gene activation [13]. DNA methylation is rare during early-stage B cells but is rather frequent in mature B cells and is regulated by three classes of methyl transferases: DNMT1, DNMT3A, DNMT3B [17]. DNMT3A and DNMT3B are responsible for de novo methylation. During transit of centroblasts through the GC, *DNMT3A* expression is markedly downregulated. In contrast, *DNMT1* is significantly upregulated, showing a central role in GC development and during replication, maintaining DNA methylation and genomic integrity as documented by the observation that its reduced expression leads to DNA damage in hypomorphic mice [18,19].

Demethylation is mediated by *TET* and occurs during early B cell stages and in the GC, operating on enhancers and in heterochromatic regions, respectively. Its central role in the maturation program was demonstrated in *TET2* and *TET3* knockout mice, where the lack of these elements blocks the transition from pro- to pre-B cells in the bone marrow, resulting in decreased expression of IRF4 and an impaired IgK rearrangement [20].

### 2.2. Histone Modifications

Histone proteins organize DNA in nucleosomes, wrapping around the histone core composed of an octamer of H2A, H2B, H3, and H4 proteins. Nucleosomes are the fundamental element of chromatin and can be present in two different conformations: heterochromatin (condensed state) and euchromatin (accessible state) [21]. The N-terminal tails of histone proteins can undergo post-translational modifications, changing the chromatin properties and consequently the gene expression. The most frequent histone modifications include acetylation, methylation, ubiquitination, phosphorylation, and sumoylation [22]. Each modification is carried out by specific enzymes that introduce or remove chemical groups from terminal histone amino acids, and the combination of these changes represents a “histone code” that will determine chromatin conformation (Figure 1) [23].

Histone acetyltransferases and histone deacetylases (HADCs) are responsible for the addition and removal of an acetyl group, respectively. Acetylation is normally associated with gene expression, and it commonly takes place at the H3K9, H3K18, H3K27, H3K4, and H3K5 lysine residues. Among these epigenetic modifications, bromodomain and extraterminal motif (BET) family proteins act as readers interacting with hyper-acetylated lysine residues and promoting gene transcription. Specifically, they can occupy super-enhancers of oncogenes such as *MYC*, *CCDN2*, and *MCL-1* [24]. Instead, histone methylation can occur on arginine, lysine, and histidine residues, is mediated by methyltransferases and the most frequently methylated sites are H3K4 (catalyzed by MLL1-5, SET1-A, SET1-B), H3K9 (SUV39-H1), H3K27 (EZH2), H3K36 (SET2, NSD1), and H3K79 (DOT1-L) [25]. Lysine and arginine can be represented in three different methylation statuses (mono-, bi-, and tri-methylated) depending on how many methyl groups are linked to terminal amino acid residues.

During T cell dependent activation, a subset of naïve B cells can be induced to migrate into lymphoid follicles, becoming centroblasts and forming the dark zone of the GC, while another fraction of naïve B cells differentiates into low-affinity plasma cells after immunoglobulin switch recombination. Centroblasts are subjected to rapid clonal expansion and somatic hypermutation mediated by activation-induced cytosine deaminase (AICDA) introducing point mutations at cytosine residues.

EZH2, a component of the polycomb repressive complex2 (PRC2), is responsible for the trimethylation of H3K27 and has a pivotal role in GC formation and antibodies production [26]. EZH2 levels are low during early B cell stages but increase at the time of GC transit. The addition of H3K27me3 at gene promoters, already marked by H3K4me3 in naïve B cells, leads to transcriptional silencing of the *IRF4* and *PRDM1* genes involved in plasma cell differentiation and of the *CDKN1A* (p21) and *CDKN1B* check point regulators of proliferation. EZH2 levels are lower in centrocytes due to the action of the KDM6A-B demethylase, restoring the active state of promoters and inducing differentiation program [27,28].

DOT1L is involved in H3K79 methylation which is associated with active transcription during B cell differentiation, promoting the expression of a pro-proliferative and pro-GC transcriptional program. In vitro studies on B cells with *DOT1L* deletion revealed an aberrant differentiation and premature acquisition of plasma cells features [29].

The transcriptional repressor BCL-6 cooperates with EZH2 in GC-cells, recruiting SMRT and NCOR (corepressor proteins) and thus binding HDAC3, which mediates H3K27 deacetylation. During this stage, the *PAX5*, *CD19*, *EBF1*, and *SPIB* genes are expressed [30]. This deacetylation effect is reversible upon CD40 signaling in the GC, promoting SMRT phosphorylation trough ERK signaling and initiating the exit of B cells from the GC [31].

Among these epigenetic modifications, BET proteins are crucial regulators of gene expression that bind acetylated lysine residues on histones. BET proteins act as epigenetic readers, recruiting the positive transcription elongation factor b (p-TEFb) and promoting gene transcription. Specifically, they can occupy super-enhancers of oncogenes such as *MYC*, *CCDN2*, *MCL-1,* and an aberrant acetylation of these sites results in cancer cell proliferation, survival, and oncogenic progression [24,32].

### 2.3. Chromatin Remodeling

The nucleosome core particle is the basic structural unit of chromatin, consisting of DNA wrapped around a core of histone proteins [33].

The arrangement of nucleosomes and their distribution across chromatin determines how DNA becomes accessible to nuclear macromolecules, which control essential genetic operations including DNA replication, transcription, recombination, and repair (Figure 1) [34,35].

B cell maturation depends on chromatin remodeling for its regulation, especially in the GC where activated B cells undergo clonal expansion, somatic hypermutation (SHM) and class-switch recombination (CSR) [36]. The regulation of these processes depends on the dynamic control of gene accessibility, which is achieved through chromatin remodeling complexes, histone modifications, and changes in 3D genome architecture [37].

The SWI/SNF (BAF) complex through its catalytic subunit BRG1 functions as a crucial chromatin remodeling complex that drives B cell activation and GC development [38]. BRG1 enables B cell proliferation and GC formation through enhancer region targeting, resulting in elevated chromatin accessibility and activation of cell cycle gene transcription. Absence of BRG1 in B cells blocks the start of GC responses while preventing proper class-switched plasma cell development [38]. B cells undergo substantial 3D chromatin rearrangements after antigen recognition to activate essential GC genes including *BCL6* and *AICDA*. The chromatin organization becomes disrupted when histone H1 mutations occur, leading to chromatin decompaction and enabling GC B cells to self-renew more frequently [39].

### 2.4. Noncoding RNA

In mature B cell neoplasms, gene expression is also controlled by noncoding RNAs that influence multiple regulatory pathways. Both long noncoding RNAs (lncRNAs) and microRNAs (miRNAs) play key roles in epigenetic mechanisms such as genomic imprinting and chromatin remodeling (Figure 1) [40]. These noncoding RNAs act as signals, guides, decoys, or scaffolds, modulating gene expression at transcriptional and post-transcriptional levels. Dysregulation of lncRNAs and miRNAs in B cell lymphomas disrupts genes involved in B cell differentiation while promoting tumorigenic gene expression patterns [41].

lncRNAs are involved in cellular differentiation and tissue development, while miRNAs are involved in B cell maturation and activation in both normal and neoplastic B cells The abnormal regulation of these noncoding RNAs leads to drug resistance in CLL and lymphomas, which makes them useful as biomarkers for diagnosis, prognosis, and therapeutic response [42]. Noncoding RNAs control gene expression during B cell maturation through their action at developmental checkpoints [43].

## 3. Epigenetic Landscape in B Cell Neoplasia

### 3.1. Diffuse Large B Cell Lymphoma

DLBCL is the most common lymphoid neoplasm in the adult population, arising from mature B cells, and it is characterized by a heterogeneous genetic and epigenetic landscape which influences morphology, immunophenotype, and therapy response. Two principal subtypes can be distinguished based on the cell of origin: the GC B cell-like (GCB) subtype, which is generally associated with a more favorable prognosis, and the activated B cell-like (ABC) subtype, which is typically linked to inferior clinical outcomes [1,2,44].

The pathogenesis of DLBCL is characterized by the accumulation of genetic aberrations causing an altered structure and expression of important proto-oncogenes and tumor suppressor genes. Genetic alterations include somatically acquired point mutations, gene copy number variation and translocations. Translocations lead to the placement of a regulatory element (promoter or enhancer), often involving IG loci, in proximity to the coding sequence of an oncogene, resulting in its constitutive expression [45]. Among the main aberrations, *MYC*, *BCL2*, and *BCL6* translocations are assessed by FISH analysis to define the double- or triple- hit status in co-presence of *MYC* and *BCL2* or/and *BCL6* alterations [46].

#### 3.1.1. DNA Methylation in DLBCL

Aberrant methylation disrupts the normal regulation of critical genes involved in cell cycle control, apoptosis, and differentiation. Among the key genes affected in DLBCL are *MYC*, *SLIT2*, *KLF4*, and *CDKN2A*, whose inactivation facilitates uncontrolled proliferation and tumor progression. DLBCL shows methylation heterogenicity which defines different molecular subgroups and clinical outcomes. In particular, six molecular subtypes have been identified based on methylation variability, with high grade methylation variability being associated with shorter survival compared to lower grade variability [47,48,49,50]. Additionally, higher methylation heterogeneity increases the probability of the appearance of DLBCL clones that are epigenetically programmed to better tolerate chemoimmunotherapy. In fact, patients with higher heterogeneity displayed a poorer outcome with a shorter survival after R-CHOP therapy [51]. Consistently, in vitro studies have demonstrated that the hypermethylation of *SMAD1*, a key transducer of TGF-β signaling, contributes to the development of chemoresistance in DLBCL cells exposed to R-CHOP treatment [52].

#### 3.1.2. Histone Modifications and Chromain Remodeling in DLBCL

Mutations in genes responsible for histone and chromatin modifications, such as *CREBBP*, *KMT2D*, *EZH2*, and *TET2*, drive aberrant epigenetic B cell programming and influence DLBCL clinical outcome [53]. Loss-of-function mutations in *CREBBP* are common in GCB-DLBCL, occur in approximately 30% of cases and lead to reduced acetyltransferase activity. Under normal conditions, CREBBP regulates enhancers that are repressed by BCL6 through cooperation with the co-repressors SMRT and NCOR in the GC [54]. This repressor complex also includes HDAC3, which mediates deacetylation of H3K27. Loss of CREBBP function results in unchecked enhancer repression, altering the expression of key genes, including those encoding MHC class II molecules [54]. Furthermore, CREBBP deficiency contributes to dysregulated B cell signaling and excessive repression of gene transcription. The CREBBP-mediated deacetylation of H3K27 downregulates the expression of FBXW7, a key negative regulator of NOTCH, and thus activates the NOTCH signaling pathway, which plays a critical role in B cell malignancies [55]. *CREBBP* mutations further disrupt the downstream components of the NOTCH pathway, leading to expression of CCL2 and CSF1, which in turn promote the polarization of tumor-associated macrophages toward the immunosuppressive M2 phenotype [56]. These alterations impair normal immune surveillance, facilitating tumor immune evasion and driving disease progression (Figure 2) [55,56].

*EZH2* mutations are found in 6–14% of DLBCL overall and in about 20% of GCB-DLBCL, where they are frequently associated with *BCL2* translocations [57]. Under normal conditions, EZH2 cooperates with BCL6 to mediate gene silencing at the promoters of genes involved in plasma cell differentiation and cell cycle regulation [28]. In mouse models, *EZH2* knockout leads to accelerated lymphomagenesis [58]. *EZH2* gain-of-function mutations, most commonly affecting the Tyr641 residue within the SET domain, drive germinal center hyperplasia and contribute to remodeling of the immune microenvironmen [57]. Unlike normal GC B cells, where cytosine methylation and H3K27me3 marks are usually mutually exclusive, in DLBCL these modifications often overlap, with hypermethylation partly affecting EZH2 target genes [59].

*KMT2D*, also known as *MLL2*, encodes a methyltransferase essential for the methylation of lysine 4 on histone H3 (H3K4), a key modification that regulates gene enhancer activity [60]. Loss-of-function mutations in *KMT2D* are found in approximately 30% of DLBCL [61]. These mutations impair H3K4 methylation, leading to the downregulation of *FBXW7*, a negative regulator of the NOTCH signaling pathway. The resulting activation of NOTCH signaling drives several oncogenic processes, including enhanced cell survival, proliferation, and inhibition of terminal B cell differentiation [62]. This aberrant signaling further promotes activation of downstream pathways such as RAS, ERK, and MYC/TGF-β1, contributing to tumor growth and immune evasion [62].

The TET family of enzymes plays a key role in active DNA demethylation, particularly at gene enhancers, where this activity is linked to transcriptional activation [63]. *TET2* mutations occur in approximately 10% of DLBCL patients and result in abnormal DNA hypermethylation at regulatory elements within the GC [64]. This hypermethylation leads to the silencing of genes involved in GC exit and B cell receptor signaling, contributing to lymphomagenesis. In *TET2* knockout mice, loss of *TET2* disrupts GC B cell homeostasis and promotes transformation into aggressive lymphomas. These models also show increased formation of G-quadruplexes and R-loops, that represent structures associated with double-strand DNA breaks at immunoglobulin switch regions [65].

#### 3.1.3. Noncoding RNA in DLBCL

miRNA dysregulation plays a key role in the pathogenesis and treatment resistance of DLBCL. High miR-21 expression has been linked to poorer survival through its suppression of FOXO1 and PTEN, activating the PI3K/AKT/mTOR pathway (Figure 2) [66]. High miR-155 levels are linked to poor response to R-CHOP therapy [67]. Overexpression of miR-125a and miR-125b targets TNFAIP3, leading to NF-κB activation and promoting a pro-proliferative and anti-apoptotic state. Similarly, increased let-7a/b/f levels reduce PRDM1 expression, impairing B cell differentiation [68]. Distinct miRNA profiles help distinguish DLBCL from other lymphomas and define disease subtypes. For example, a 27-miRNA signature differentiates Burkitt lymphoma from DLBCL, while panels including miR-28-5p, miR-214-5p, miR-339-3p, and miR-5586-5p are associated with better outcomes [69]. Conversely, reduced expression of miR-370-3p, miR-381-3p, and miR-409-3p in relapsed cases suggests their role in modulating chemosensitivity. Altogether, these findings highlight how miRNA dysregulation disrupts key signaling pathways and drives abnormal cell proliferation, survival, and differentiation in DLBCL.

### 3.2. Follicular Lymphoma

FL represents the second most prevalent form of lymphoma occurring in adults [70]. FL originates from B cells with the t(14;18) translocation, which triggers BCL2 overexpression but is not sufficient for full transformation [71]. Additional genetic mutations accumulate as these cells persist and cycle through the GC. Crucially, many of these mutations affect epigenetic regulators, leading to widespread changes in chromatin structure, histone modification, and DNA methylation. These epigenetic alterations reshape gene expression and the tumor microenvironment, driving FL development and progression. Understanding these epigenetic mechanisms is essential for identifying new therapeutic targets in FL [72,73,74,75].

#### 3.2.1. Histone Modifications and Chromatin Remodeling in FL

Mutations affecting epigenetic-related genes primarily involve histone methyltransferases like *KMT2D* and *EZH2*, as well as histone acetyltransferases such as *CREBBP* and *EP300*. Most of these mutations are inactivating, except those in *EZH2*. These genes are essential for regulating gene expression through the modification of H3K4 by KMT2D and H3K27 by EZH2, CREBBP and EP300 (Figure 2) [8,76].

*KMT2D* is the most frequently mutated gene in FL, and its loss of function contributes significantly to lymphomagenesis [61]. Loss-of-function mutations in *KMT2D* significantly influence the transcriptional regulation of immune-related genes and facilitate FL development by modifying H3K4 methylation and the expression of genes associated with B cell activation pathways, which include CD40, JAK-STAT, and B cell receptor signaling [77]. *KMT2D* mutations compromise global H3K4 methylation in GC B cells, thereby promoting proliferation and lymphomagenesis [61]. Gain-of-function mutations in *EZH2* occur in approximately 20–25% of FL cases and were among the first epigenetic alterations identified as critical drivers in the early stages of this lymphoma [27]. These mutations enhance EZH2-mediated trimethylation of H3K27 (H3K27me3), leading to abnormal repression of gene transcription. As a key epigenetic writer active in GC B cells, EZH2 silences regulators of B cell differentiation through H3K27me3 deposition. This results in the silencing of genes involved in cell cycle checkpoints, differentiation, and immune synapse signaling, reducing the dependency of GC B cells on T cell help. Consequently, lymphoma cells evade immune control while promoting their survival and proliferation. *EZH2* mutations also shift GC B cell reliance from T cell interactions to increased engagement with follicular dendritic cells, expanding the centrocyte-like population characterized by heightened proliferation and survival [27,78,79].

CREBBP functions as a haploinsufficient tumor suppressor and plays a critical role in both the development and progression of FL [80]. It regulates the expression of genes involved in B cell receptor and CD40 signaling pathways, which are essential for normal B cell activation and differentiation. Inactivating mutations in *CREBBP*, particularly those affecting its acetyltransferase (KAT) domain, disrupt chromatin modification, leading to abnormal gene expression [56]. These mutations are common in FL and can often be detected years before clinical diagnosis, highlighting their role in early disease development [81].

#### 3.2.2. DNA Methylation in FL

Although DNA methyltransferase genes are not mutated per se in FL, aberrant DNA methylation is a hallmark of the disease [82]. For instance, mutations in H1 linker histone genes impair the binding of DNMT3B, contributing to abnormal methylation patterns [83]. Promoter hypermethylation, particularly at *PRC2* target genes, leads to silencing of key tumor suppressors such as *CDKN2A*, allowing unchecked cellular proliferation [84]. On the other hand, global hypomethylation contributes to genomic instability and activation of oncogenes, further fueling tumorigenesis [85]. A more deregulated DNA methylation landscape has been associated with greater disease aggressiveness. While methylation patterns generally remain stable between pre- and post-transformation FL, relapsed FL often displays increased methylation levels [86].

Mutations of genes encoding subunits of chromatin remodeling complexes such as ARID1A, which are part of the SWI/SNF (BAF) complex, are present in a fraction of FL [72,87]. Mutations in *ARID1A* result in reduced chromatin accessibility, and loss of ARID1A expression has been associated with a shift towards a memory B cell-like state associated with increased risk of progression to aggressive lymphoma [71,87].

Recent advances underscore the role of epigenetic alterations in improving risk stratification in FL. While the traditional follicular lymphoma international prognostic index (FLIPI) score often overestimates the proportion of high-risk patients at diagnosis, the integration of genetic data in the m7-FLIPI model incorporating mutations in seven genes, including key epigenetic regulators such as *EZH2*, *CREBBP*, *EP300*, *MEF2B*, and *ARID1A*, has significantly improved prognostic accuracy [88]. This model effectively distinguishes between high- and low-risk patients and predicts progression within 24 months of first-line R-CHOP therapy, a critical surrogate for overall survival. Notably, mutations in *CREBBP* and *EP300* are associated with high-risk disease, whereas *EZH2* and *MEF2B* mutations correlate with more favorable outcomes. These findings underscore the potential of epigenetic modifier mutations to enhance risk prediction and guide treatment intensity [88].

#### 3.2.3. Noncoding RNA in FL

ncRNAs function as essential elements in the progression and formation of FL. The expression of long noncoding RNAs (lncRNAs) shows the greatest variation between FL classes, with grade 3B displaying the highest levels of unregulated lncRNAs that modulate proliferation and cell cycle [89]. FL also involves miRNAs, with 133 miRNAs showing significant differences between follicular hyperplasia and FL [90]. A unique FL miRNA profile comprises miRNAs associated with improved response to chemotherapy. MiR-20a/b and miR-194, respectively, target CDKN1A and SOCS2, which may affect the survival and proliferation of tumor cells [90]. Both lncRNAs and miRNAs affect cell proliferation, tumor response, and other biological processes in B cell development and lymphomagenesis [91,92]. Musilova and colleagues analyzed miRna profiles in serial biopsies of transformed follicular lymphoma and identified five miRnas enriched during transformation, including miR-150 [93]. Their study revealed a c-MYC/miR-150/FOXP1 regulatory axis, in which c-MYC overexpression suppresses miR-150, resulting in increased FOXP1 expression. FOXP1, a TF essential for B cell development, is associated with the ABC subtype of DLBCL and is linked to poor clinical outcomes in both DLBCL and FL [93].

### 3.3. Chronic Lymphocytic Leukemia

CLL is a mature B cell malignancy characterized by the clonal expansion and accumulation of CD5^+^CD19^+^ B lymphocytes in the blood, lymphoid organs, and bone marrow [1]. It is the most common leukemia in adults, typically following an indolent course but exhibiting significant clinical variability [94]. Advances in targeted therapies, including inhibitors of Bruton tyrosine kinase (BTK) and B cell lymphoma 2 (BCL2), have significantly improved progression-free and overall survival, particularly in patients with high-risk genetic features [95,96]. Despite therapeutic progress, the molecular mechanisms underlying CLL pathogenesis remain only partially understood. Increasing evidence suggests that epigenetic alterations play a central role in shaping disease onset, progression, and response to therapy, highlighting the need for a deeper insight into the epigenetic landscape of CLL.

#### 3.3.1. DNA Methylation in CLL

CLL is characterized by widespread alterations in DNA methylation, most notably global hypomethylation relative to healthy B cells [97]. Interestingly, the global DNA methylation landscape in CLL is remarkably stable over time and largely conserved between circulating resting cells and those in proliferative lymphoid niches [98]. This epigenetic stability across disease compartments supports the notion that aberrant DNA methylation is established early in leukemogenesis and may serve as a foundational event in CLL pathobiology (Figure 2).

Distinct DNA methylation patterns among CLL subtypes reflect epigenetic imprints inherited from their respective cells of origin, influencing both the biological behavior of leukemic cells and the clinical course of the disease [97,99]. Mutated IGHV CLL (M-CLL) retain a DNA methylation pattern resembling that of normal post-GC (memory-like) B cells, whereas unmutated CLL (U-CLL) exhibit a naïve-like methylation signature [100]. Interestingly, these analyses have also identified a third epigenetic subtype of CLL with an intermediate methylation profile, falling between the naïve-like and memory-like patterns. This suggests that the intermediate subtype may arise from a yet unidentified normal B cell subset. These three epigenetically defined CLL subtypes exhibit distinct patterns in terms of somatic mutation landscapes, immunoglobulin heavy chain variable region (IGHV) gene usage, and clinical outcomes [97,100,101]. Naïve-like CLL typically associates with unmutated IGHV genes, frequent usage of IGHV1-69, 11q deletions, 2p16 gains and recurrent *NOTCH1* mutations. The intermediate subtype often displays partially mutated IGHV genes, with a higher frequency of IGHV3-21, IGHV1-18, and BCR stereotyped subset #2, alongside elevated rates of *SF3B1* and *MYD88* mutations. In contrast, memory-like CLL is characterized by mutated IGHV genes, commonly utilizing IGHV4-34 and IGHV3-7 [101]. Importantly, these epigenetic subtypes also correlate with significant differences in time to first treatment and OS [100]. The prognostic significance of this epigenetic classification has been validated in independent studies [100,102,103]. Collectively, these findings support the notion that the cell of origin, defined by its immunogenetic or epigenetic signature, plays a crucial role in shaping the biology and clinical behavior of CLL.

The expression of tumor suppressor genes (TSGs) is frequently silenced through promoter hypermethylation in CLL. The Wnt signaling pathway, which plays a crucial role in normal B cell development and regulation of apoptosis, is often aberrantly activated in CLL [104]. One of the key negative regulators of this pathway, secreted frizzled-related protein 4 (SFRP4), is a member of the SFRP family and is commonly hypermethylated in CLL samples [105]. Hypermethylation of CpG islands in the promoters of *SFRP* genes, including *SFRP4*, leads to transcriptional silencing and may contribute to the pathological activation of Wnt signaling in CLL [106]. Among the five SFRP family members, SFRP1 was found to be consistently hypermethylated and downregulated in CLL, suggesting that its epigenetic inactivation is a critical event in leukemogenesis.

Whole-epigenome analyses tracking DNA methylation dynamics throughout the course of treatment have revealed that specific epigenetic features in CLL may serve as predictors of clinical outcomes [107]. Notably, epigenetic silencing of *HOXA4* has been associated with reduced therapeutic sensitivity, indicating its potential role in chemoresistance [107]. In one study, ibrutinib treatment induced significant epigenetic and transcriptomic remodeling. Chromatin accessibility shifted toward a more closed state overall, with increased accessibility at ERK-related TF sites and reduced accessibility at BCR-related sites [108]. Most differentially expressed genes were downregulated, including those linked to IL-4, NF-κB, metabolism, and proliferation, while MAPK and RAS signaling genes were upregulated [108]. These results underscore the critical role of epigenetic reprogramming in shaping therapeutic response beyond genetic evolution alone. Further studies are needed to elucidate the role of epigenetic changes in resistance to other targeted agents, including BCL2 inhibitors as well as covalent and non-covalent BTK inhibitors.

A recent analysis of de novo chromatin activation in CLL revealed two distinct and opposing chromatin activation programs: one associated with progressive disease (PD) and the other with indolent disease (ID) [109]. De novo chromatin activation refers to the aberrant opening and activation of chromatin regions that are normally inactive in healthy B cells, enabling inappropriate gene expression that contributes to leukemogenesis [110]. These two programs correspond to underlying molecular pathways that drive disease behavior and progression. A balance score, quantifying the relative activation of PD and ID chromatin signatures, was shown to outperform conventional markers such as IGHV mutation status as an independent predictor of time to first treatment. Notably, the PD signature is enriched for TNF-α/NF-κB and mTOR signaling pathways and is shaped by the lymph node microenvironment, whereas the ID signature appears largely independent of microenvironmental influences. These findings underscore the critical role of epigenetic characterization in advancing our understanding of CLL biology and refining risk stratification.

#### 3.3.2. Histone Modifications and Chromatin Remodeling in CLL

Beyond aberrant DNA methylation, CLL exhibits widespread epigenetic remodeling that includes substantial changes in chromatin accessibility and histone modification patterns [111,112]. While mutations in epigenetic regulators are less frequent compared to other B cell lymphomas, namely FL and DLBCL, alterations in chromatin modification nonetheless play a significant role in CLL pathogenesis [113].

Although some aspects of the CLL chromatin landscape appear to be shaped by epigenetic dynamics inherited from normal B cell maturation, integrative chromatin state analyses have uncovered approximately 500 de novo activated regulatory elements common to CLL cells regardless of their underlying genetic heterogeneity. These elements correspond to enhancers or promoters of genes with established roles in CLL pathogenesis, including *TCF4*, *FMOD*, *CTLA4*, and *LEF1* [110,114]. Interestingly, the activation of these regulatory regions appears to be orchestrated by a limited number of TF families, notably NFAT, FOX, and TCF/LEF, whose binding motifs are enriched across CLL-specific chromatin-accessible sites [99,110,114]. Further investigation into the mechanistic roles of these TFs may elucidate their contribution to disease initiation and persistence, and potentially uncover therapeutic vulnerabilities related to aberrant chromatin activation.

Complementary to enhancer-focused studies, additional epigenomic analyses have revealed an unexpected co-occurrence of histone modifications in CLL that are typically considered mutually exclusive in normal cells. This epigenetic anomaly suggests the presence of intra-tumoral heterogeneity at the chromatin level, likely reflecting a mixture of subpopulations with distinct cellular identities and epigenetic states. As with DNA methylation, such chromatin diversity may provide valuable insights into subclonal evolution and disease progression in CLL.

Among the histone-modifying enzymes implicated in CLL is EZH2. Although recurrent *EZH2* mutations are common in FL and DLBCL, CLL is distinct in exhibiting overexpression of *EZH2* without accompanying genetic alterations [101]. Papakonstantinou et al. demonstrated that *EZH2* is overexpressed at both the mRNA and protein levels in a subset of patients with U-CLL, and that *EZH2* overexpression correlates with elevated global H3K27me3 levels [115]. Interestingly, U-CLL cases with high *EZH2* expression show a lower incidence of *TP53* mutations, suggesting a distinct molecular subset within the disease [115]. In vitro studies showed that overexpression of *EZH2* enhances leukemic cell viability, while *EZH2* knockdown or pharmacological inhibition induces apoptosis, supporting its role as an oncogenic epigenetic regulator [115]. Treatment with EZH2 inhibitors reduces H3K27me3 levels and triggers apoptotic responses in vitro, providing a potential therapeutic angle for targeted intervention.

Analogous to FL and DLBCL, mutations in histone-modifying enzymes are also observed in CLL, though at lower frequency. Alterations in genes such as *SETD2*, *ARID1A*, and *CHD2* further emphasize the pivotal role of epigenetic dysregulation in CLL pathogenesis [101,116,117]. *SETD2*, a histone methyltransferase responsible for H3K36 trimethylation, is mutated or deleted in up to 4% of cases and is associated with impaired genomic stability, chemoresistance, and poor clinical outcomes, often co-occurring with *TP53* alterations [117,118]. Similarly, loss-of-function mutations in *ARID1A*, a core component of the SWI/SNF chromatin remodeling complex, may drive proliferation through dysregulation of key cell cycle genes [101]. *CHD2*, part of the CHD family of chromatin remodelers, is mutated in approximately 5% of cases, especially in mutated-IGHV CLL [116]. Collectively, these findings underscore the broader relevance of histone modification and chromatin remodeling pathways in CLL biology and prognosis, supporting further exploration of epigenetic vulnerabilities as therapeutic targets (Figure 2).

#### 3.3.3. Noncoding RNA in CLL

One of the most frequently observed genetic alterations in CLL is the deletion of the miR-15a/16-1 cluster located at chromosome 13q14, occurring in more than 65% of cases and resulting in the downregulation of miR-15a/16-1 [119,120]. Notably, the first nine nucleotides of the 5′ ends of miR-15/16 are complementary to bases 3287–3279 of the 3′ UTR of BCL2 mRNA, an anti-apoptotic gene found to be overexpressed in nearly all CLL patients. Conversely, in normal physiology, the direct interaction between miR-15/16 and BCL2 leads to repression of BCL2 expression and promotion of apoptosis.

Moreover, miR-34b/c, located on chromosome 11q, functions as a tumor suppressor by negatively regulating protein translation [121]. Its expression is frequently reduced due to promoter hypermethylation, a change that strongly correlates with 11q deletions and contributes to aggressive CLL phenotypes [122]. Similarly, miR-155, encoded within the MIR155HG (BIC) locus, is a conserved microRNA that modulates B cell gene expression at the post-transcriptional level [123,124,125]. Overexpression of miR-155 enhances BCR signaling responsiveness and is often driven by microenvironmental cues. Recent studies show that *XPO1* mutations in CLL further contribute to this dysregulation by increasing chromatin accessibility and transcription of MIR155HG and its regulator MYB, leading to elevated miR-155 levels [126]. This upregulation amplifies BCR signaling, promotes CLL cell proliferation, and is associated with shorter time to first treatment (TTFT), independent of IGHV status. Together, these findings underscore the critical roles of miR-34b/c loss and miR-155 overexpression in driving aggressive disease and highlight their potential as therapeutic targets in CLL [126].

Several other miRNAs have also been implicated in CLL pathogenesis. For instance, miR-29 has demonstrated anti-tumor activity by targeting TRAF4, and its expression is modulated by BCR inhibitors, making it a potential therapeutic target [127]. miR-125a-5p and miR-34a-5p have emerged as predictive markers for Richter transformation [128]. miR-150, which is abundantly expressed in CLL cells, has been linked to disease progression through its regulation of GAB1 and FOXP1 [129].

## 4. Therapeutic Strategies Targeting Epigenetic Alterations

Over the past decade, our understanding of the epigenetic landscape in B cell malignancies has expanded significantly, revealing how aberrant DNA methylation, histone modifications, and dysregulated chromatin architecture contribute to malignant transformation, disease progression, and therapy resistance. These reversible changes offer promising therapeutic targets, enabling modulation of gene expression without modifying the underlying DNA sequence. A summary of the ongoing clinical trials investigating epigenetic drugs (epi-drugs) in mature B cell NHL is presented in Table 1.

### 4.1. Demethylating Agents

DNMT inhibitors such as 5-azacytidine and decitabine, initially developed for AML, have shown limited efficacy as monotherapy in lymphoid malignancies like DLBCL, FL, and CLL. Clinical trials using these agents alone in B cell malignancies have been largely disappointing, prompting a shift toward combination strategies [130,131,132]. While combinations with agents like HDAC inhibitors (e.g., romidepsin) have demonstrated some success in peripheral T-cell lymphoma (PTCL), other regimens, such as azacytidine with immune checkpoint inhibitors, have failed to show sufficient clinical activity [133,134]. Efforts to enhance efficacy through co-administration with enzyme cytidine deaminase (CDA) inhibitors or novel DNMTis are ongoing, but robust clinical benefits in B cell malignancies remain unproven (Figure 3).

### 4.2. HMT Inhibitors

The development of HMT inhibitors in B cell malignancies has concentrated on targeting EZH2. Among EZH2 inhibitors, tazemetostat has emerged as the most clinically advanced. Preclinical studies showed that tazemetostat was effective in vitro and in vivo (including in xenograft models) against *EZH2*-mutant lymphoma. In a phase I trial, tazemetostat demonstrated a favorable safety profile and achieved a 38% overall response rate (ORR) in refractory B-NHL [135]. A subsequent phase II trial in relapsed/refractory (R/R) FL confirmed higher activity in EZH2-mutant patients (ORR 69%) compared to *EZH2*-wild-type patients (ORR 35%) [136]. These data led the FDA to grant tazemetostat accelerated approval for the treatment of R/R FL with *EZH2* mutations. In addition, tazemetostat was evaluated in a small DLBCL cohort in combination with R-CHOP, achieving an ORR of 86% [137]. However, when combined with the PD-L1 inhibitor atezolizumab in R/R DLBCL, no added benefit was observed (Figure 3) [138]. This highlights the ongoing challenge of improving outcomes through combination therapies. Further trials are exploring new combinations and strategies to enhance the efficacy of tazemetostat and other HMT inhibitors, aiming to improve overall survival in patients with these lymphomas.

Other agents have shown more limited efficacy. For instance, GSK126, despite its strong preclinical activity against mutant *EZH2*, failed to show meaningful clinical responses in a phase I trial in R/R DLBCL, transformed FL, and other NHL, leading to early trial termination [139,140]. Valemetostat, a dual EZH1/2 inhibitor, has shown promise in early-phase studies, achieving an ORR of 53% in B- and T-NHL, with particularly high activity (80% ORR) in T-cell lymphoma [141].

### 4.3. HDACs Inhibitors

HDACs play a key role in regulating cell proliferation, growth, and angiogenesis, which are often dysregulated in cancer. Overexpression of HDACs promotes tumor development, making HDAC inhibitors (HDACis) a promising anticancer strategy (Figure 3) [142]. HDACis induce apoptosis, cause cell cycle arrest, and can modulate immune responses, leading to therapeutic effects in various lymphomas [143]. Vorinostat, the first FDA-approved HDACi (2006), targets class I and II HDACs and is approved for relapsed/refractory cutaneous T-cell lymphoma (CTCL) [144]. It has demonstrated benefits as monotherapy in FL and in combination regimens with rituximab, fludarabine, or etoposide for relapsed/refractory B-NHL, CLL, and T-NHL. Belinostat, a pan-HDACi, is approved for relapsed/refractory PTCL and has shown activity in CTCL, although with limited efficacy in B-cell lymphomas. Romidepsin, a class I HDACi, is also approved for relapsed/refractory CTCL and PTCL and has demonstrated additive or synergistic effects when combined with gemcitabine, 5-azacitidine, or alisertib in T-cell lymphomas and FL [145]. Mocetinostat showed limited efficacy as monotherapy in relapsed CLL, with most patients tolerating only two treatment cycles, and no responses observed when combined with rituximab [146]. Ricolinostat, a selective HDAC6 inhibitor, exhibited disease stabilization in ~50% of patients with relapsed/refractory B- and T-cell lymphomas in early clinical studies, though no objective responses were recorded; it also showed synergistic effects in preclinical DLBCL models when combined with agents like ibrutinib or carfilzomib [147,148,149]. Fimepinostat, a dual HDAC/PI3K inhibitor, achieved response rates of 64% in *MYC*-altered DLBCL versus 37% in non-*MYC*-altered cases when combined with rituximab. Panobinostat showed modest activity in relapsed/refractory DLBCL and limited benefits in Hodgkin lymphoma trials; combination strategies yielded mixed efficacy and notable toxicities [148,150]. Abexinostat, another pan-HDACi, demonstrated promising results in phase II trials, with overall response rates ranging from 31% to 56% in FL, PTCL, and DLBCL. These findings underscore the potential of HDACis in lymphoma therapy, particularly in biomarker-selected or combination treatment contexts [151].

### 4.4. BET Inhibitor

Early BET inhibitors (BETi) showed promise preclinically, but clinical outcomes have been mixed. Birabresib, the first BETi tested in R/R DLBCL, achieved a CR rate of 47% in 17 patients, though only 18% had durable OR (Figure 3) [152]. Other BETis, including INCB054329 and INCB057643, demonstrated minimal clinical activity with no meaningful responses and significant toxicity, leading to early trial termination. CPI-0610 showed a modest 7% overall response rate (ORR) in B-NHL [153]. A combination of the BETi RO6870810 with venetoclax (± rituximab) in R/R DLBCL achieved an ORR of 38.5%, including 20.5% CR and 17.9% partial response (PR), with stable disease in 15.4% of patients, but without clear evidence of synergistic benefits [154]. These data underscore the challenges in translating BET inhibition into durable clinical responses in B-cell malignancies.

## 5. Discussion and Conclusions

The last decade has witnessed remarkable progress in our understanding of the epigenetic underpinnings of mature B cell neoplasms. Epigenetic dysregulation including aberrant DNA methylation, histone modifications, chromatin remodeling, and noncoding RNA activity emerges not only as a hallmark of tumor biology but also as a key driver of disease heterogeneity, immune evasion, and therapeutic resistance across DLBCL, FL, and CLL. Mutations in chromatin-modifying enzymes such as *EZH2*, *KMT2D*, *CREBBP*, and *TET2* among others, disrupt normal gene expression programs essential for B cell maturation and facilitate malignant transformation. Beyond genetic lesions, many of these epigenetic alterations converge on common oncogenic pathways, leading to the upregulation or downregulation of BCR signaling, NF-κB activity, and apoptosis regulators. The activity of these pathways, whether altered through epigenetic mechanisms or co-existing mutations, can define aggressive disease subsets and influence therapeutic response, underscoring the need to account for their status when designing epigenetic-targeted strategies. Furthermore, the interplay between epigenetic modifiers and transcription factors, along with the contribution of noncoding RNAs, defines disease-specific regulatory networks with clinical relevance. Although current therapeutic strategies targeting the epigenome have shown promise in selected patient subsets, their clinical impact remains variable and often limited as monotherapy. Emerging evidence underscores the crucial role of epigenetic mechanisms in shaping the tumor microenvironment by modulating immune-related processes and fostering immune evasion. Targeting these epigenetic alterations may offer a promising strategy to reprogram the TME and enhance the efficacy of novel immunotherapies (i.e., chimeric antigen receptor (CAR) T-cells and bispecific antibodies) through synergistic combination approaches. These challenges underscore the necessity of developing rational combination regimens, better biomarkers of response, and subtype-specific approaches guided by the molecular and epigenetic context of each neoplasm. The advent of integrative multi-omics and single-cell epigenetic profiling is poised to unravel intra-tumoral heterogeneity and inform precision medicine strategies. While this review focuses on DLBCL, FL, and CLL as representative mature B-cell neoplasms, epigenomic alterations are also critical in other subtypes. In mantle cell lymphoma, mutations in key regulators such as *KMT2D*, *NSD2*, *SMARCA4*, *SP140*, and *KMT2C* contribute to disease biology, with epigenomic profiling distinguishing SOX11-positive and SOX11-negative variants of differing aggressiveness [155]. In Hodgkin lymphoma, alterations in *BCL6* and *CREBBP*, along with EBV-driven epigenomic modulation, play major pathogenic roles [156,157]. Ultimately, a deeper mechanistic understanding of the epigenetic architecture governing B cell malignancies holds promising potential, not only for refining prognostic models and improving risk stratification but also for unlocking new therapeutic avenues that can overcome resistance and induce durable remissions. The convergence of epigenetic insight and clinical innovation stands to redefine the therapeutic landscape for patients with mature B cell neoplasms.

## Figures and Tables

**Figure 1 ijms-26-08132-f001:**
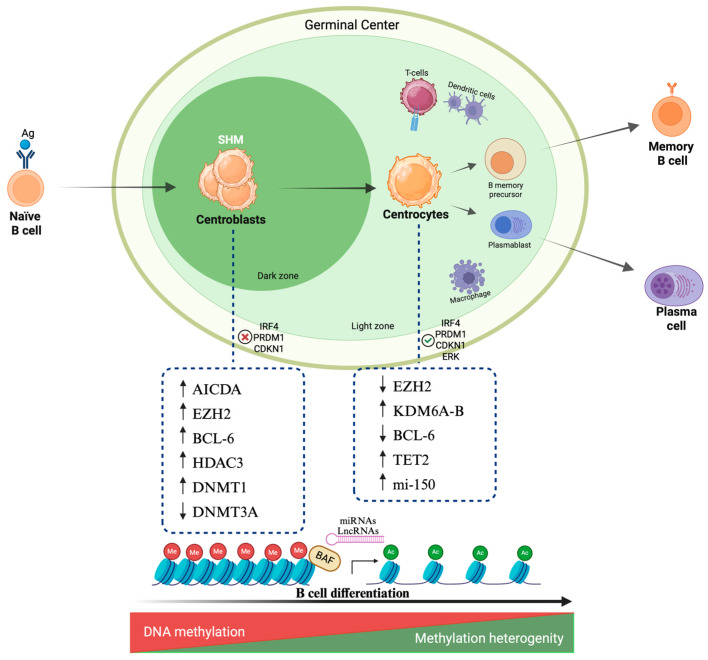
**Epigenetic overview of B cell maturation.** The main epigenetic regulators include DNA methylation, histone modifications, chromatin remodeling, and lnc-RNAs. The interplay among these four modifications regulates gene expression and shapes the molecular features that are specific to each cell subtype during B cell maturation. In centroblasts, DNA methylation mediated by DNMT1 is frequent while DNMT3 expression is downregulated. During this stage, AICDA is responsible for SHM introducing point mutations at cytosine residues. EZH2 is upregulated, inducing histone hypermethylation, silencing IRF4, PRMD1 (genes involved in plasma cells differentiation) and decreasing CDKN1levels. High levels of BCL-6 allow deacetylation modulated by HDAC3. In centrocytes, EZH2 levels decrease, resulting in transcription of genes involved in cell differentiation. DNA methylation levels are heterogenous among later stages of differentiation, reflecting heterogeneity of cell differentiation. Acetylation mediated by HAC promotes gene expression regulated by BAF complex and BET proteins. Increased mi-150 levels in late stage of maturation are necessary to complete B cell differentiation and GC exit. Created in BioRender.com. Gaidano G. (2025) https://BioRender.com/ddw3e9n (accessed on 19 August 2025). *Lnc-RNA*, long noncoding RNA; *DNMT*, DNA methyltransferase; *AICDA*, activation induced cytidine deaminase; *SHM*, Somatic Hypermutation; *Ac*, acetylation; Me: methylation.

**Figure 2 ijms-26-08132-f002:**
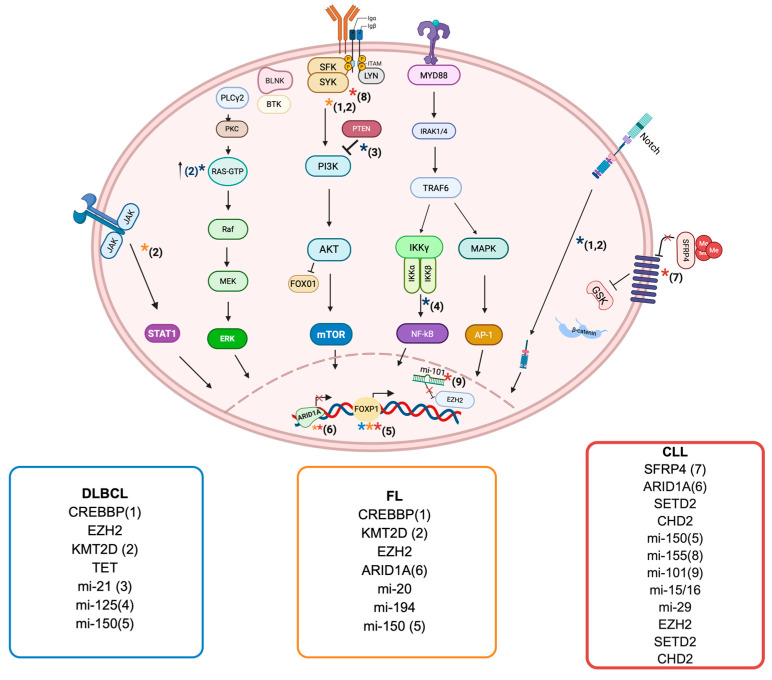
**Mutations and main epigenetic regulators in B cell malignancies.** The most common epigenetic regulators and their mutations are listed in colored boxes representing the most frequent mature B cell malignancies: (blue = DLBCL; orange = FL; red = CLL). Arrows indicate the functional consequences of mutations or deregulated non-coding RNAs on signaling pathways. Colored stars define the disease context in which each alteration occurs. Numbers identify the genes and the epigenetic regulators involved. Their impact on different cell pathways is reported by colored stars which define the disease; numbers identify the genes and the epigenetic regulators involved. CREBBP loss of function mutations result in overactivation of NOTCH and BCR pathways in DLBCL and FL. KMT2D mutations significantly impact on JAK-STAT and BCR signaling in FL. RAS-ERK and MYC pathways are activated in DLBCL. Increased mi-21 and mi-125 levels activate mTOR and NF-kB pathways in DLBCL, thus promoting proliferation. Mi-150 promotes FOXP1 expression, mediating B cell transformation in DLBCL, FL and CLL. ARID1A mutations result in increased tumor progression in FL and CLL. Hypermethylation of SFRP1 induces its own downregulation as well as the hyperactivation of WNT signaling in CLL. Increased mi-155 levels enhance BCR pathway, mi-101 downregulation leads to EZH2 overexpression and deletion of mi-15/16 increases BCL-2 expression promoting survival in CLL cells. Created in BioRender.com. Gaidano G. (2025) https://BioRender.com/nufxqjt (accessed on 19 August 2025). *DLBCL*, diffuse large B cell lymphoma; *FL*, follicular lymphoma; *CLL*, chronic lymphocytic leukemia.

**Figure 3 ijms-26-08132-f003:**
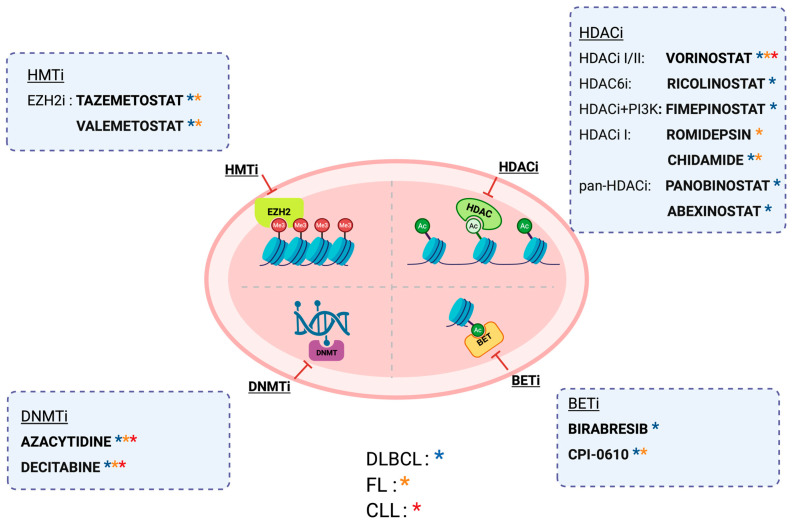
**Therapeutic strategies for epigenetic alterations in B cell malignancies.** The four main epigenetic modifications and their inhibitors are represented in the figure. Colored asterisks identify the diseases for which the drug is approved, or studies that are ongoing. (blue = DLBCL; orange = FL; red = CLL). Created in BioRender.com. Gaidano G. (2025) https://BioRender.com/nv3vzvq (accessed on 19 August 2025). *HMTi*, histone methyltransferase inhibitor; *HDACi*, histone deacetylase inhibitor; *DNMTi*, DNA methyltransferase inhibitor; *BETi*, bromodomain and extraterminal motif inhibitors.

**Table 1 ijms-26-08132-t001:** Ongoing clinical trials of epigenetic drugs in DLBCL, FL, and CLL.

Drug	Epigeneitc Target	Phase	Setting	Disease	Primary Outcome	Identifier
Decitabine + anti PD-1	DNMT	Phase II	R/R	DLBCL	ORR	NCT05816746
ASTX727 + Nivolumab	DNMT	Phase I	R/R	NHL and HL	Safety	NCT05272384
Tazemetostat + Mosunetuzumab	EZH2	Phase II	First-line	FL	CR	NCT05994235
Tulmimetostat DZR123 (CPI-0209)	EZH2	Phase I/II	R/R	B and T lymphomas, and solid tumors	Safety ORR	NCT04104776
Tazemetostat + CART	EZH2	Phase I	R/R	DLBCL, FL, and MCL	Safety	NCT05934838
tazemetostat + lenalidomide + rituximab	EZH2	Phase I/II	R/R	FL	Safety PFS	NCT04224493
Mevrometostat (PF-06821497)	EZH2	Phase I	R/R	FL	Safety	NCT03460977
Mevrometostat	EZH2	Phase I	R/R	FL, SCLC, and CRPC	Safety	NCT03460977
Tazemetostat + Belinostat	EZH2 + HDACs	Phase I	R/R	B-NHL and T-NHL	Safety	NCT05627245
HH2853	EZH2	Phase I/II	R/R	DLBCL, and FL	Safety ORR	NCT04390737
SHR2554 + SHR1701	EZH2	Phase I/II	R/R	B-cell lymphoma, and Advanced/metastatic solid tumors	PFS	NCT04407741
Chidamide + Decitabine	HDACs + DNMT1	Phase I/II	R/R	NHL after CART cells	CR AE	NCT04337606
Chidamide + Linperlisib	HDACs + PI3Kδ	Phase Ⅱ	R/R	FL	CR	NCT06158386
Purinostat	HDAC	Phase II	R/R	DLBCL	ORR	NCT05563844
Abexinostat	HDAC	Phase II	R/R	DLBCL	ORR	NCT03936153
Etinostat + ZEN003694	HDAC + BET	Phase I/II	R/R	Lymphoma, and advanced solid tumors	Safety ORR	NCT05053971
Vorinostat + pembrolizumab	HDAC	Phase I	R/R	DLBCL, FL, and HL	Safety	NCT03150329
Tazemetostat + epcoritamab	EZH2	Phase II	R/R	FL	Safety CR	NCT06575686
Abexinostat	HDAC	Phase II	R/R	FL	CR or PR	NCT03600441
Valemetostat + Lenalidomide	EZH1/EZH2	Phase I/II	R/R	FL	Safety	NCT05683171
Valemetostat	EZH1/EZH2	Phase II	R/R	B-cell lymphoma	ORR	NCT04842877
Pevoneidstat + Ibrutinib	NEDD8-activating enzyme	Phase I	R/R	CLL + B-NHL	Safety	NCT03479268
Tulmimetostat	EZH1/EZH2	Phase I/II	R/R	Solid tumors + Lymphoma	Safety ORR	NCT04104776

Abbreviations: *DNMT*, DNA methyl transferase; *R/R*, relapsed/refractory; *DLBCL*, diffuse large B cell lymphoma; *ORR*, overall response rate; *NHL*, non-Hodgkin lymphoma; *HL*, Hodgkin lymphoma; *EZH2*, enhancer of zeste homolog 2; *FL*, follicular lymphoma; *CR*, complete remission; *CART*, chimeric antigene receptor T cells; *MCL*, mantle cell lymphoma; *SCLC*, small cell lung cancer; *CRPR* castration resistant prostate cancer; *HDAC*, histone deacetylase; *BET*, bromodomain and extraterminal motif.

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
