# Peer review of "Unraveling the Epigenetic Landscape of Mature B Cell Neoplasia: Mechanisms, Biomarkers, and Therapeutic Opportunities"

_ijms, 2025, doi:10.3390/ijms26178132_

Round 1
Reviewer 1 Report
Comments and Suggestions for Authors
The current manuscript provides an overview of the epigenetic alterations and the relevant targeted therapies for mature B-cell neoplasia, including DLBCL, FL, and CLL. The review is interesting and potentially informative for clinical practice and the development of therapeutic options. However, the review in the current format should further address the following concerns.
1) A number of review articles regarding epigenetic alterations in B-cell cancers have been published. The current focus is only on three types of mature B cell neoplasia. The novelty and comprehensiveness of the current format are significant concerns that should be addressed in the introduction section.
2) The definition of mature B-cell neoplasia is not clearly described. This is important because a significant part of the manuscript focuses on the development of B cells. A relevant question is how the cancers differ from the mature cells in their cellular immune phenotypes. Are the epigenetic findings related to the phenotypes?
3) Although the mutations directly associated with epigenetic regulations were described, many other mutations, such as BCR signaling (BTK, CD79), NF-kB pathways, and apoptosis (BCL2, TP53), were not. A key question is whether these mutations are associated with disease progression. These pathways or signaling hyperactivation are more prominent in some subsets of patients. Furthermore, when targeting epigenetic alterations, treatment options and their efficacies should take these factors into account. The authors should consider integrating these factors.
4) In discussing the mature B cell neoplasia, only three, including DLBCL, FL, and CLL, are relatively more common. However, multiple myeloma is also very common, particularly in the elderly, which, together with other forms, should be discussed as well.
5) In Table 1, the title and table only include clinical trials and drugs for “B-NHL”. CLL is most common, for which drugs and trials should be included if there are any.
6) The information needs to be updated in Table 1. For example, Valemetostat, a dual inhibitor for EZH1/EZH2, should be included. Abexinostat (PCI-24781), a HDAC inhibitor tested for CLL/Fl should be considered.
7) Minor issues: Lines 144 and 176, the abbreviation for BET should be explained when it first appears in the text of the manuscript.
8) The section numbers need to be consistent. For example, Diffuse Large B Cell Lymphoma (DLBCL) is assigned in Section 3.1, but Follicular Lymphoma (Line 331) starts with the number Section 3.5.
Author Response
The current manuscript provides an overview of the epigenetic alterations and the relevant targeted therapies for mature B-cell neoplasia, including DLBCL, FL, and CLL. The review is interesting and potentially informative for clinical practice and the development of therapeutic options. However, the review in the current format should further address the following concerns.
Comments 1: A number of review articles regarding epigenetic alterations in B-cell cancers have been published. The current focus is only on three types of mature B cell neoplasia. The novelty and comprehensiveness of the current format are significant concerns that should be addressed in the introduction section.
Response 1: We thank the Reviewer for her/his constructive comments. In the revised introduction, we have explicitly highlighted the novelty and comprehensiveness of our review by emphasizing the focused analysis of epigenetic alterations across three specific mature B-cell neoplasms. We also clarify how this targeted approach provides deeper insights that complement existing broader reviews, thereby addressing both the uniqueness and the scope of our work (see lines 75-85).
Comments 2: The definition of mature B-cell neoplasia is not clearly described. This is important because a significant part of the manuscript focuses on the development of B cells. A relevant question is how the cancers differ from the mature cells in their cellular immune phenotypes. Are the epigenetic findings related to the phenotypes?
Response 2: We thank the Reviewer for the comment. We have added a brief definition of mature B-cell neoplasms and clarified how epigenetic changes contribute to their altered immune phenotypes (see lines 36-46).
Comments 3: Although the mutations directly associated with epigenetic regulations were described, many other mutations, such as BCR signaling (BTK, CD79), NF-kB pathways, and apoptosis (BCL2, TP53), were not. A key question is whether these mutations are associated with disease progression. These pathways or signaling hyperactivation are more prominent in some subsets of patients. Furthermore, when targeting epigenetic alterations, treatment options and their efficacies should take these factors into account. The authors should consider integrating these factors.
Response 3: BCR signaling, NF-κB, and apoptosis pathways are already discussed in the manuscript as common downstream targets modulated by multiple epigenetic alterations, through either upregulation or downregulation, rather than mutations alone (see lines 332–334, 484-486, 486–488, 501–504, 558-564, 570-576, 579–581) and are illustrated in Figure 2. We have now clarified this in the Discussion (see lines 687-693), noting that altered activity of these pathways can define aggressive disease subsets and should be considered when designing epigenetic-targeted therapies, as they may influence progression and treatment response.
Comments 4: In discussing the mature B cell neoplasia, only three, including DLBCL, FL, and CLL, are relatively more common. However, multiple myeloma is also very common, particularly in the elderly, which, together with other forms, should be discussed as well.
Response 4: While we acknowledge that multiple myeloma is a common mature B-cell neoplasm, we have chosen to focus this review on DLBCL, FL, and CLL due to their shared lymphoid origin, overlapping epigenetic mechanisms, and the availability of comparable epigenomic data. We have now clarified this rationale in the Introduction (see lines 75-79).
Comments 5: In Table 1, the title and table only include clinical trials and drugs for “B-NHL”. CLL is most common, for which drugs and trials should be included if there are any.
Response 5: The title of Table 1 has been modified to specifically reflect its focus on DLBCL, FL, and CLL.
Comments 6: The information needs to be updated in Table 1. For example, Valemetostat, a dual inhibitor for EZH1/EZH2, should be included. Abexinostat (PCI-24781), a HDAC inhibitor tested for CLL/Fl should be considered.
Response 6: We have updated Table 1 to include additional epigenetic agents.
Comments 7: Minor issues: Lines 144 and 176, the abbreviation for BET should be explained when it first appears in the text of the manuscript.
Response 7: The abbreviation for BET (Bromodomain and Extraterminal Domain) has now been defined at its first occurrence in the manuscript (lines 156-157).
Comments 8: The section numbers need to be consistent. For example, Diffuse Large B Cell Lymphoma (DLBCL) is assigned in Section 3.1, but Follicular Lymphoma (Line 331) starts with the number Section 3.5.
Response 8: The section numbering has been reviewed and corrected to ensure consistency throughout the manuscript.
Reviewer 2 Report
Comments and Suggestions for Authors
This manuscript delivers a highly comprehensive, well-structured, and up-to-date review of the epigenetic mechanisms underpinning the development, progression, and therapeutic vulnerabilities of mature B cell neoplasms, including DLBCL, FL, and CLL. It synthesizes a vast body of literature spanning molecular biology, immuno-oncology, and translational therapeutics. The authors demonstrate outstanding command of the field, presenting both conceptual and clinical insights with clarity and depth.
Author Response
Comments 1: This manuscript delivers a highly comprehensive, well-structured, and up-to-date review of the epigenetic mechanisms underpinning the development, progression, and therapeutic vulnerabilities of mature B cell neoplasms, including DLBCL, FL, and CLL. It synthesizes a vast body of literature spanning molecular biology, immuno-oncology, and translational therapeutics. The authors demonstrate outstanding command of the field, presenting both conceptual and clinical insights with clarity and depth.
Response 1:
We thank the Reviewer for her/his positive and encouraging feedback
Reviewer 3 Report
Comments and Suggestions for Authors
The authors wrote a clear and concise review on the epigenetic dysregulations (i.e., aberrant DNA methylation events, histone modifications, dysregulated chromatin architecture, etc.) that are associated with in mature B cell neoplasms (with a focus on the epigenetic dysregulations occurring in diffuse large B cell lymphoma, follicular lymphoma, and chronic lymphocytic leukemia). The authors review and discuss the mechanisms responsible for these epigenic dysregulations, the genes affected, and the opportunities for pharmacologic intervention in the form of targeted therapies that are currently used to treat mature B cell neoplasms. A number of ongoing clinical trials designed to evaluate DNMT, HDAC, HMT, and BET inhibitors in patients afflicted with these conditions are mentioned while the efficacies of various epigenetic drugs and drug combinations are also discussed. I found the review to be not only very informative, but also well written and clearly structured, which I am sure will benefit the potential readers interest in this topic. I recommend the current version of the manuscript to be considered for publication by the Editors.
A very minor comment: the PRDM1 gene on line 163 is misspelled and needs to be corrected.
Author Response
Comments 1:
The authors wrote a clear and concise review on the epigenetic dysregulations (i.e., aberrant DNA methylation events, histone modifications, dysregulated chromatin architecture, etc.) that are associated with in mature B cell neoplasms (with a focus on the epigenetic dysregulations occurring in diffuse large B cell lymphoma, follicular lymphoma, and chronic lymphocytic leukemia). The authors review and discuss the mechanisms responsible for these epigenic dysregulations, the genes affected, and the opportunities for pharmacologic intervention in the form of targeted therapies that are currently used to treat mature B cell neoplasms. A number of ongoing clinical trials designed to evaluate DNMT, HDAC, HMT, and BET inhibitors in patients afflicted with these conditions are mentioned while the efficacies of various epigenetic drugs and drug combinations are also discussed. I found the review to be not only very informative, but also well written and clearly structured, which I am sure will benefit the potential readers interest in this topic. I recommend the current version of the manuscript to be considered for publication by the Editors.
A very minor comment: the PRDM1 gene on line 163 is misspelled and needs to be corrected.
Response 1:
We thank the Reviewer for her/his positive and encouraging feedback. We have corrected the minor typo related to PRDM1 on line 176 as suggested.
Reviewer 4 Report
Comments and Suggestions for Authors
The manuscript by Maher et al. presents a well-written, well-conceptualized, and thoroughly referenced review of epigenetic regulation in mature B cell neoplasms, with a focus on diffuse large B-cell lymphoma (DLBCL), follicular lymphoma (FL), and chronic lymphocytic leukemia (CLL). The figures are nice, clear, and illustrative, and the table is informative and well-designed.
In the abstract, the authors state: “Epigenetic regulation is critical to B cell development, guiding gene expression via DNA methylation, histone modifications, chromatin remodeling, and noncoding RNAs. In mature B cell neoplasms, particularly diffuse large B cell lymphoma (DLBCL), follicular lymphoma (FL), and chronic lymphocytic leukemia (CLL), these mechanisms are frequently disrupted.” This theme is thoroughly explored in the main text, where the epigenetics of these three malignancies are reviewed in detail.
Epigenetic dysregulation also plays a significant role in the pathogenesis and clinical heterogeneity of other mature B cell neoplasms, such as mantle cell lymphoma (MCL), which demonstrates a wide range of clinical behaviors - from indolent to highly aggressive, and Hodgkin lymphoma (HL), which is characterized by a unique phenotype and loss of typical B-cell features, likely linked to epigenetic changes. Therefore, it would be helpful to clarify the rationale for focusing exclusively on DLBCL, FL, and CLL.
Was the selection based on their relative prevalence (DLBCL as the most common aggressive B-NHL, FL as the most common indolent NHL, CLL as the most common leukemia in adults in Western countries), the relative abundance of available epigenetic data, their therapeutic relevance, or because MCL and HL have been recently reviewed elsewhere? A brief explanation of this selection criterion in the introduction or discussion would help readers better understand the scope and focus of the review. Additionally, a brief acknowledgment of key epigenetic findings in MCL and HL could further enrich the manuscript and provide a more comprehensive overview of the role of epigenetics across mature B cell malignancies.
Author Response
Comments 1:
The manuscript by Maher et al. presents a well-written, well-conceptualized, and thoroughly referenced review of epigenetic regulation in mature B cell neoplasms, with a focus on diffuse large B-cell lymphoma (DLBCL), follicular lymphoma (FL), and chronic lymphocytic leukemia (CLL). The figures are nice, clear, and illustrative, and the table is informative and well-designed.
In the abstract, the authors state: “Epigenetic regulation is critical to B cell development, guiding gene expression via DNA methylation, histone modifications, chromatin remodeling, and noncoding RNAs. In mature B cell neoplasms, particularly diffuse large B cell lymphoma (DLBCL), follicular lymphoma (FL), and chronic lymphocytic leukemia (CLL), these mechanisms are frequently disrupted.” This theme is thoroughly explored in the main text, where the epigenetics of these three malignancies are reviewed in detail.
Epigenetic dysregulation also plays a significant role in the pathogenesis and clinical heterogeneity of other mature B cell neoplasms, such as mantle cell lymphoma (MCL), which demonstrates a wide range of clinical behaviors - from indolent to highly aggressive, and Hodgkin lymphoma (HL), which is characterized by a unique phenotype and loss of typical B-cell features, likely linked to epigenetic changes. Therefore, it would be helpful to clarify the rationale for focusing exclusively on DLBCL, FL, and CLL.
Was the selection based on their relative prevalence (DLBCL as the most common aggressive B-NHL, FL as the most common indolent NHL, CLL as the most common leukemia in adults in Western countries), the relative abundance of available epigenetic data, their therapeutic relevance, or because MCL and HL have been recently reviewed elsewhere? A brief explanation of this selection criterion in the introduction or discussion would help readers better understand the scope and focus of the review. Additionally, a brief acknowledgment of key epigenetic findings in MCL and HL could further enrich the manuscript and provide a more comprehensive overview of the role of epigenetics across mature B cell malignancies.
Response 1:
We thank the Reviewer for her/his positive and encouraging feedback. While several reviews focus on individual B-cell malignancies or specific epigenetic pathways, our review aims to provide a comprehensive and integrated analysis across three of the most prevalent mature B-cell lymphoid neoplasms, namely DLBCL, FL, and CLL. These were selected due to their shared lymphoid origin, overlapping epigenetic features, and the availability of comparable epigenomic data. We have now clarified this rationale in the Introduction section (see lines 36-45 and 75-79). In addition, to address the Reviewer’s request, a few sentences on the importance of epigenetics also in MCL and HL have been added to the Discussion (see lines 706-713).
Reviewer 5 Report
Comments and Suggestions for Authors
This is a very interesting manuscript regarding the role of epigenetic control in lymphomas. The first part reviews the physiologic role of this control. Thereafeter the authors describe the role of this disregulation in lymphomas and the potential use, as therapeutical targets.
Comments (minor).
- Authors should describe the physiological role of IDH1 & IDH2. The describe the role of the mutations of these enzymes, most in the adjuvant role with TET2 family during lymphomagenesis, but the physiological role is not described.
Author Response
Comments 1:
This is a very interesting manuscript regarding the role of epigenetic control in lymphomas. The first part reviews the physiologic role of this control. Thereafeter the authors describe the role of this disregulation in lymphomas and the potential use, as therapeutical targets.
Comments (minor).
Authors should describe the physiological role of IDH1 & IDH2. The describe the role of the mutations of these enzymes, most in the adjuvant role with TET2 family during lymphomagenesis, but the physiological role is not described.
Response 1:
We thank the Reviewer for her/his comment. Information regarding IDH1/2 role have been added (see lines 119-123).
Round 2
Reviewer 4 Report
Comments and Suggestions for Authors
I have no further comments